# Ethical Decision-Making in Law Enforcement: A Scoping Review

Ronald P. Dempsey [1] , Elizabeth E. Eskander [1,2] and Veljko Dubljević [1,]*

1 Department of Philosophy and Religious Studies, North Carolina State University, Raleigh, NC 27695-8103, USA
2 Department of Psychology, North Carolina State University, Raleigh, NC 27695-8102, USA
* Correspondence: veljko_dubljevic@ncsu.edu

**Abstract:** Decision-making in uncertain and stressful environments combined with the high-profile cases of police violence in the United States has generated substantial debates about policing and created challenges to maintaining public confidence and trust in law enforcement. However, despite the manifestations of reactions across the ideological spectrum, it is unclear what information is available in the literature about the convergence between ethical decision-making and policing. Therefore, an interdisciplinary scoping review was conducted to map the nature and extent of research evidence, identify existing gaps in knowledge, and discuss future implications for ethical decision-making in law enforcement. This review investigates the interaction between the job complexities of policing (psychological and normative factors) and aspects of ethical decision-making, synthesizing three distinct themes: (1) socio-moral dimensions impact the job complexities of police work, (2) lethal means and moral injury influence intuitive and rational decision-making, and (3) police wellness and interventions are critical to sustaining police readiness. Gaps in recruiting, training, and leadership and managerial practices can be broadly transformed to fundamentally emphasize officer wellness and a holistic approach to ethical practices, enabling police officers to uphold the rule of law, promote public safety, and protect the communities they serve.

**Keywords:** law enforcement; police; ethics; decision-making; sociology; moral psychology; interventions; policy; scoping review





## 1. Introduction

Recent high-profile cases of police violence in the United States have generated substantial controversy and increased research surrounding the role of police, including their ethical practices and decision-making, lack of accountability, and racial disparities in deaths by law enforcement [1–4]. Moreover, these incidents continue to exacerbate the tenuous relationship law enforcement has with the public. For instance, the tragic killing of Michael Brown in Ferguson, Missouri, in 2014 brought into focus the underlying problems of excessive use of force by law enforcement and instances of police brutality, sparking numerous public protests and dissatisfaction with the police [5]. Further compounding the public perception of policing is the underreporting of fatal police violence and the declaration of police violence as an urgent public health crisis [3,6].

As noted above, the criticism of the ethics of policing has only grown in volume in recent years. Nonetheless, police officers who serve the public in a free society must have the ability to uphold every aspect of their profession to promote public safety and maintain the rule of law [7]. Doing so requires a sense of ethics and the ability to use moral reasoning to navigate the ambiguous job complexities of law enforcement and the life-and-death circumstances police may encounter at a moment's notice. While it has always been clear that moral integrity has a strong correlation with upholding the law, law enforcement agencies only started adopting ethics training in the last quarter of the 20th century despite

a long evolution [8]. Although ethics is part of the curriculum in police academies, the level and depth of training varies [9], and it is easy to conceive of the adopted ethics training as merely a checklist. Ethical codes of conduct are still relatively new, and by and large, it is uncontroversial to assert that there is not enough substance of ethics either in police academies or in the actual practice of law enforcement. Furthermore, when combined with acts of police violence, and society's response and calls for reform, ethical decision-making in uncertain and stressful environments present an emerging challenge to maintaining the confidence and trust of the public for law enforcement professionals [9,10].

### 1.1. Rationale

It is clear that police officers, in the course of their duties, make many split-second value-based decisions that affect the lives and well-being of countless individuals and groups. Policing is also ethically evaluated by society and the specific communities that law enforcement agencies serve. The job complexities of policing, and high rates of fatalities among law enforcement officers, have drastically complicated the right and wrong of decisions and behaviors, especially in terms of perceiving everyone as a potential threat. Thus, a subculture of policing has emerged that engrains an us versus them mentality and potentially provides a false dichotomy to fall back upon (i.e., criminals are bad, and it is the job of police officers to protect the community from these criminals). The tough-on-crime perspective is conducive to ethical and moral decisions that reflect the need to combat crime and punish criminals, not treat them respectfully. From this perspective, one could argue that stopping criminals and treating them harshly is the right behavior and not unethical. For much of what the police do, the grey area of policing, right and wrong is not so clear. For example, when arresting a parent for illegal substance use, knowing the children are in the home and observing the encounter, an officer may view this as the most ethical decision, while the community might view this as unethical due to the negative impact this arrest may have on the children's mental and physical well-being [10]. Thus, in addition to the controversy surrounding police violence, public confidence may be affected by gaps in training, less than optimal participation in community engagement, the appearance of corrupt or unethical policing cultures, or a lack of understanding of policing practices. Nevertheless, it is unclear what types of information are available in the extant literature about the convergence between ethics and policing.

For these reasons, a scoping review was undertaken to map the research in this area and identify gaps in knowledge, suggesting future directions for research in the field. We carefully documented our findings through a descriptive, quantitative presentation of the results. Next, the findings are expanded upon by discussing theoretical and conceptual implications, and then we synthesize the identified research to facilitate the development of policies and programmatic efforts for ethical decision-making in policing. Furthermore, we emphasize how including ethical theories and models in this study may enhance the results, making them more useful to policymakers, law enforcement practitioners, and the general public who desire to understand the job complexities of police work.

### 1.2. Objectives

This scoping review aims to report on the state of decision-making approaches in policing. Additionally, the review seeks to examine whether there are different kinds of ethical decision-making methodologies, the kinds of low- and high-stakes dilemmas for which decision-making approaches have been used, how normative and psychological factors influence moral judgment in policing, and what facilitators or barriers have been reported relating to the methodologies' successes and failures.

We applied normative ethics when formulating the research questions and to guide the scoping review to assess how the existing literature incorporates ethical theories in their evaluation of ethical decision-making in policing. Normative ethics is the branch of philosophy that studies moral conceptions of right, value, and moral worth and focuses on the major theoretical approaches to moral structure, grammar, and evaluation [11,12].

More generally, ethics can be described as a set of norms, rules, precepts, and principles that govern and influence the behaviors of individuals or groups of people and are of salience to understanding decision-making in policing. Furthermore, research has shown that although there is evidence that has been corroborated for specific kinds of moral theory, there is some empirical evidence that virtue ethics, deontology, and consequentialism are ordinarily present in moral judgment [11,13]. Accordingly, the following questions guided this scoping review, as shown in Table 1.

**Table 1.** Research questions.

| Research Question | Background | References |
|---|---|---|
| What are the socio-moral dimensions of policing? | Deontology is concerned with a person's (police officer's) actions (use of force or use of deception), claiming that specific actions are either right or wrong based on the intent of the actions. | [1–6,11–13] |
| What are the virtues of law enforcement agents or police officers? | Virtue ethics focus on the agency or character of a person performing actions, attributing differences in moral character as the argument for why people (police officers) act differently in identical situations. | [14] |
| What are the anticipated outcomes when investigating the moral aspects of decision-making in policing? | Consequentialism emphasizes the outcomes of actions and argues that an agent (police officer) is moral if it only chooses the most ethical outcome after weighing its relative moral value. | [14] |
| What are the key factors of unfamiliar micro-dilemmas involving split-second decisions and the use of force and deception in policing? | Research shows there are numerous factors that motivate law enforcement, use of force, and use and use of deception. Interdisciplinary application of sociology and psychology to include moral psychology may facilitate multi-level factor analysis and uncover how normative and psychological factors (job complexities) influence moral judgment in policing. | [13,15–17] |
| To what extent can holistic moral judgment models such as the agent, deed, and consequence (ADC) model facilitate understanding moral evaluations in policing? * | The integrative model facilitates the creation of a coding schema to qualitatively analyze and synthesize an understanding of moral evaluations in policing. | [11,15] |
| What gaps exist in law enforcement training and policy? | Supplementary literature outside of scoping review sample suggests that there are training and policy gaps. | [8,9] |

* The agent, deed, and consequence (ADC) model states that moral judgment consists of three different components: the character of a person (agent), their actions (deed), and the consequences brought about by the situation (consequence) [11,15]. The ADC model applies virtue ethics, deontology, and consequentialism moral theories to these components.

## 2. Materials and Methods

We performed a scoping review following the research and reporting methods established and published by Tricco and colleagues that aligns with the Preferred Reporting Items for Systematic Reviews and Meta-Analyses for Scoping Reviews (PRISMA-ScR) methodology [18]. The scoping review was not registered, but the a priori proposal was reviewed by an interdisciplinary panel of academic experts. The stages of the scoping review included (1) establishing the research team, (2) formulating the rationale, objectives, and research questions, (3) identifying, screening, and selecting relevant literature and studies, (4) data extraction, coding, collating, and summarizing, and (5) reporting the results.

The research team developed standards and protocols for database searches using Covidence. Covidence is an internet-based systematic workflow platform that facilitates collaborative research for different types of review studies. For more information, see https://www.covidence.org/ (accessed on 1 September 2021). The initial search strategies were drafted by R.D. and E.E. and further refined through team discussion. These strategies were applied to two different databases: Web of Science and APA Psych Info. In addition,

both databases were accessed in September 2021 for research published in academic journals with no constraint placed on the publication date. The search was conducted using the following search terms and related terms for Web of Science and APA Psych Info: police OR law enforcement (All Fields) AND moral OR ethic* (All Fields) AND decision* (All Fields).

After developing our search strategy, we adopted the population, concept, and context (PCC) tool to formulate inclusion and exclusion criteria. The inclusion criteria restricted inclusion to only those sources of evidence that included law enforcement officers as the population or part of a population. However, it is essential to note that law enforcement officers were excluded from the population if they fell outside the roles of law enforcement officers at the local and state levels when viewed in the context of law enforcement in the United States. For instance, studies that solely focused on customs or border agents were excluded because of the distinct differences in how they execute their duties and responsibilities. This exclusion parameter did not apply to examining evidence of law enforcement populations in other countries to account for variance in law enforcement approaches to ethical decision-making outside of the United States. The core concept examined by the scoping review was decision-making in the context of ethics. Therefore, any studies that deviated or fell outside the parameters of the PCC framework were excluded.

Two authors independently screened titles, abstracts, and full texts for the articles identified from the database search to determine which articles would be selected for further assessment. A reliability rating of 92.2% was established for the title and abstract screening, and a reliability rating of 85.9% was established for the full-text screening to introduce rigor into the process of including and excluding studies in the review. Disagreements were resolved by discussion and input from the third author. A unanimous agreement was reached regarding which articles to include in the current review.

### 2.1. Selection of Studies

The studies that did not meet the design requirements were excluded. As seen in Figure 1, 277 records were extracted for screening. Twenty duplicate records were removed from the initial screening. Of the 257 records that remained, 186 were deemed outside the scope of this review during the title and abstract screening based on various eligibility requirements such as the population, concept, context, peer-reviewed, and language accessibility. Seventy-one (71) studies were selected for full-text screening; of those remaining, 17 were removed due to the following reasons: wrong study population ($n = 4$), wrong study population and concept ($n = 2$), wrong concept ($n = 3$), missing article body or information ($n = 2$), and not peer-reviewed ($n = 6$). The types of evidence to be included in the scoping review were peer-reviewed articles consisting of primary research studies and reviews to reduce the ambiguity in navigating the abundance of literature published on law enforcement. Evidence such as opinion papers, law notes, media publications, and letters would not be particularly appropriate or useful to explore ethical decision-making due to increased bias regarding the profession, ideological influence, and experiences with policing. After this final screening, 54 studies were included in the scoping review, as shown in Figure 1, PRISMA-ScR Flowchart (see the Appendix A, Table A1 for a list of the articles used in the scoping review).

### 2.2. Data Extraction, Management, and Analysis

A customized data extraction method was developed to explore the scope of the available literature and compare the study design and methodology and manage the results of the 54 included studies (see Table 2). The data collected were entered into a spreadsheet, which was made available to the review team to manage the results of the 54 studies. Initially, we considered numerous dimensions for data analysis, but data were coded based on the abductive inference approach to qualitative research [19]. The abductive inference approach is distinctly different from both deduction and induction but combines features

of both types of inference. The goal is to iteratively cultivate new observations starting with predefined data, which seeks to advance the understanding of ethical decision-making in policing and account for job complexities and other sociological factors. Intercoder reliability was established at 85.42% using a pilot sample of 11 articles.

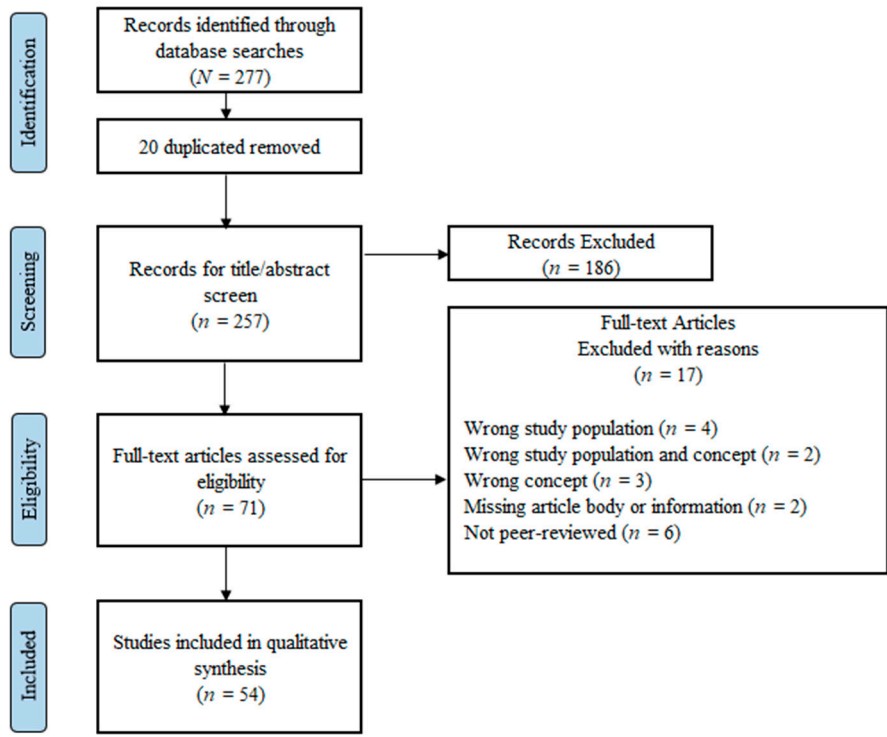

**Figure 1.** Preferred Reporting Items for Systematic Reviews and Meta-Analyses for Scoping Reviews (PRISMA-ScR) Flowchart.

**Table 2.** Detailed study characteristics.

| Detailed Study Method Characteristics [1] | Study Sample *N* = 54 | |
|---|---|---|
| | **Frequency** | **Percentage** |
| **Qualitative Methods** | *n* = **13** | **24.07%** |
| Semi-structured interviews | *n* = 3 | 5.55% |
| Ethnographic analysis | *n* = 1 | 1.85% |
| Scoping review | *n* = 1 | 1.85% |
| **Quantitative Methods** | *n* = **25** | **46.30%** |
| Surveys | *n* = 6 | 11.11% |
| Questionnaires | *n* = 5 | 9.25% |
| Content analysis | *n* = 4 | 7.40% |
| Longitudinal study | *n* = 1 | 1.85% |
| Moral judgment test | *n* = 1 | 1.85% |
| **Mixed Methods** | *n* = **6** | **11.11%** |
| **Conceptual** [2] | *n* = **10** | **18.52%** |

[1] The studies examined in this scoping review varied in their research approach. The primary methods, and corresponding frequency and percentage are bolded to represent the broad research approaches—adding up to 100 percent of the sample. [2] Conceptual research involves investigation and explanation of new ideas and theories not dependent on empirical findings.

## 3. Results

The results were grouped into the primary domains of normative ethics, normative/structural effects, and psychological dimensions and were further broken down into categories and codes (see the Appendix A, Table A2 for an overview of the domains,

main categories, and codes used for the qualitative analyses in this scoping review). This approach facilitated the development of a thematic framework to identify and map the emerging themes, including gaps and interventions in the 54 studies as illustrated in Figure 2, scoping review themes, domains, and categories. Additionally, the 54 studies examined varied in their research methodologies, frequency of yearly publication since 1980, as shown in Table 2, detailed study characteristics, and scoping review articles' publication frequency over time, as shown in Figure 3.

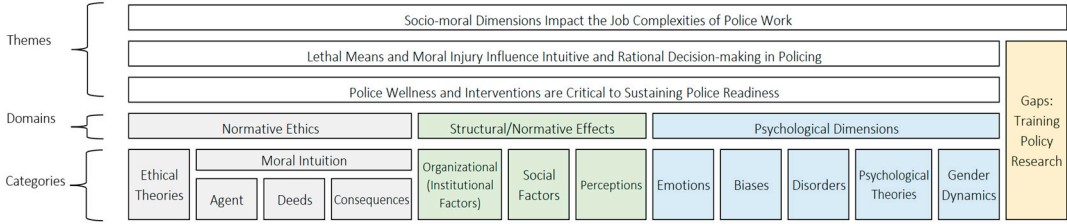

**Figure 2.** This scoping review synthesized three distinct themes: (1) socio-moral dimensions impact the job complexities of police work, (2) lethal means and moral injury influence intuitive and rational decision-making, and (3) police wellness and interventions are critical to sustaining police readiness.

The literature sample includes one article published during the 1980s, and from the start of the 1990s through 2013, yearly publications became more frequent. As shown in Figure 2, the majority of peer-reviewed literature in this scoping review increased significantly after 2013, with seven articles published in 2019. This frequency distribution most likely coincides with increased public scrutiny and social media consumption, bringing attention to the highly controversial instances of police violence in the United States [4,17,20–22]. Additional future research and analysis could confirm that a frequency distribution relationship exists, but it is not in the scope of this review's objectives.

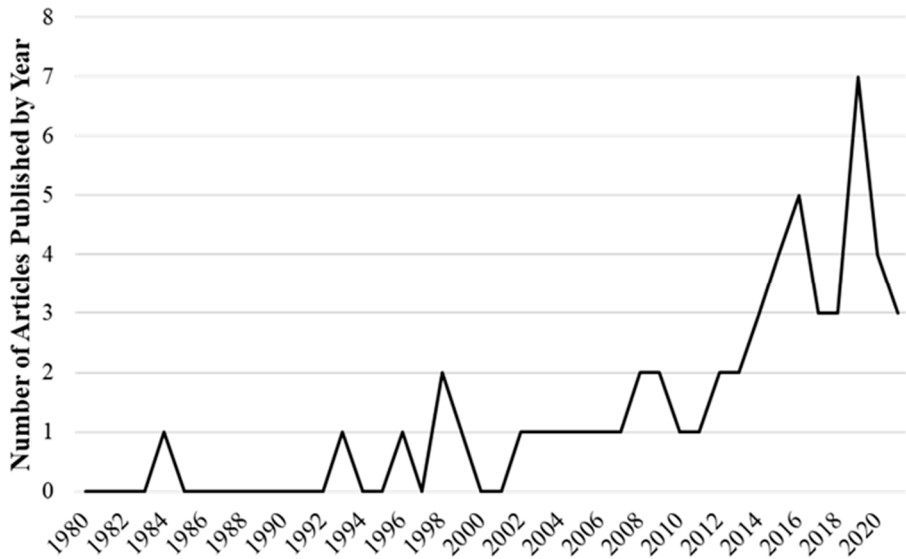

**Figure 3.** Scoping review articles' publication frequency over time.

### 3.1. Normative Ethics

Of the 54 reviewed articles, one-third discussed at least one of the three ethical theories, with virtue ethics being the most prominent and deontology and consequentialism receiving less consideration from researchers (see Figure 4, ethical theory representation across the sample). In comparison, nearly one-third of the literature sample made no mention of any ethical theories. Additionally, a small number of articles discussed all three ethical theories in their studies. Articles making explicit and implicit mention of virtue ethics,

deontology, and consequentialism constituted the remainder of the literature sample. There was no mention of virtue ethics and consequentialism discussed explicitly or implicitly together in an article in the literature sample. Separately, the moral theory known as the ethics of care (*n* = 5, 9%) appeared in the literature sample and, in one instance, examined care-based moral reasoning in the context of the ethic of care and justice development, which has the potential to have an impact on ethical decision-making [23]. The following excerpts highlight several prominent occurrences of implicit discussions of normative ethical theories from the scoping review sample.

> <u>**Virtual Ethics**</u> **(implicit):** *"Immediately upon their recruitment to the police academy, trainees are instilled with foundational values such as integrity, citizenship, justice, and pride [ . . . ]. Police recruits are trained in a way that prioritizes serving and protecting civilians in the community, even at their peril."* [24] (p. 71)

> <u>**Deontology**</u> **(implicit):** *"Despite the moral conflicts experienced in the line of duty, police officers are mandated to make decisions and fulfill their responsibilities in a way that is compatible with police values, and more broadly, with society's values regarding morally acceptable behaviour. Police officers are asked to do what is "right" and to maintain peace and order. When police officers feel that they have not satisfied this mandate, they may experience moral struggles that, in turn, may have a number of negative outcomes, such as increased vulnerability to stress, adverse reactions to traumatic incidents, and poorer job performance [ . . . ]."* [24] (p. 72)

> <u>**Consequentialism**</u> **(implicit):** *"An attitude toward whistleblowing [ . . . ] is the sum of the products of the employee's beliefs about the consequences of whistleblowing and his or her subjective evaluation of those consequences."* [25] (p. 546)

The data identifies a wide disparity in the presentation of theories and raises questions about the kind of theories and goals that should guide ethical decision-making for police officers. Moreover, the costs of favoring one theory and neglecting inclusion of the others could create gaps in understanding the socio-moral complexities of police work and risk in orientating policies and interventions.

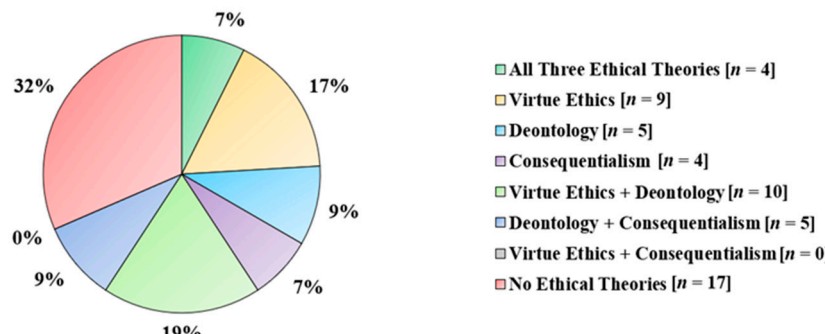

**Figure 4.** Ethical theory representation across the sample.

The domain of normative ethics also focuses on moral intuition and utilizes the agent, deed, and consequence (ADC) model to help illuminate the socio-moral dimensions of policing, what the virtues of law enforcement agents or police officers are, and what the anticipated outcomes of decision-making in policing are. The model states that moral judgment consists of three different components: the character of a person (agent), their actions (deed), and the consequences brought about by the situation (consequence) [11]. This model is useful for capturing a variety of situations and dimensions due to its ability to determine the three overarching domains of ethical concerns and their application to law enforcement, as shown in Figure 5, count of agent nodes in the literature sample, Figure 6, count of deed nodes in the literature sample, and Figure 7, count of consequence nodes in

the literature sample. See Table 3: agent, deed, and consequence domains with nodes and characteristic quotes from the sample for illustrative code excerpts.

The most common agent nodes were integrity (33%), fairness, (28%), ability (advantageous and disadvantageous) (22%), and loyalty (22%). The virtues of integrity and fairness in following the contours of strong moral principles were consistently used in the context of police officers morally doing the correct thing. In contrast, lack of integrity (corruption) (9%) was mentioned somewhat less frequently. Research emphasis was placed on understanding the organizational dynamics and job complexities that shape police integrity and the application of fairness as justice [14,25–27]. Subsequently, these dynamics create an ecology where other values such as loyalty, as a function of survival affecting moral choices [28], take precedence over integrity in the course of police work, potentially creating undesirable behaviors (deeds) and unfavorable outcomes (consequences) [29]. The value of honesty (13%) was mentioned in parallel with integrity, but it is surprisingly limited in the literature sample with an emphasis on the importance honesty plays in the recruitment and hiring of law enforcement officers [30,31]. Other virtues and values, such as compassion (7%) and empathy (7%), were mentioned less frequently in the literature sample.

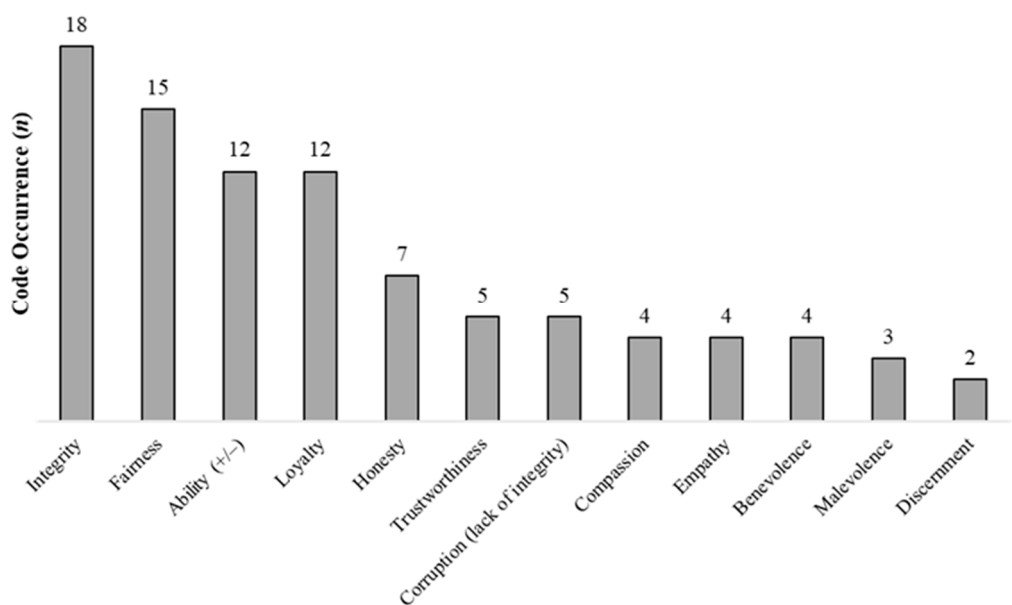

**Figure 5.** Count of agent nodes in the literature sample.

The most common deed nodes were the use of force (31%), use of discretion (28%), misconduct (28%), and preventing distress and harm (22%). Other deeds such as the use of authority (20%), use of procedural justice (19%), use of deception (17%), and whistleblowing (15%) were mentioned less frequently in the literature sample. The articles in the literature sample used coercion (17%) to capture the use of force and threats underneath one umbrella term. In the scoping review, it is essential to note that procedural justice and whistleblowing deeds (actions) have gained increased scholarly attention as an indicator of countering deviant police subcultures, mitigating misconduct, and supporting ethical behaviors and decision-making.

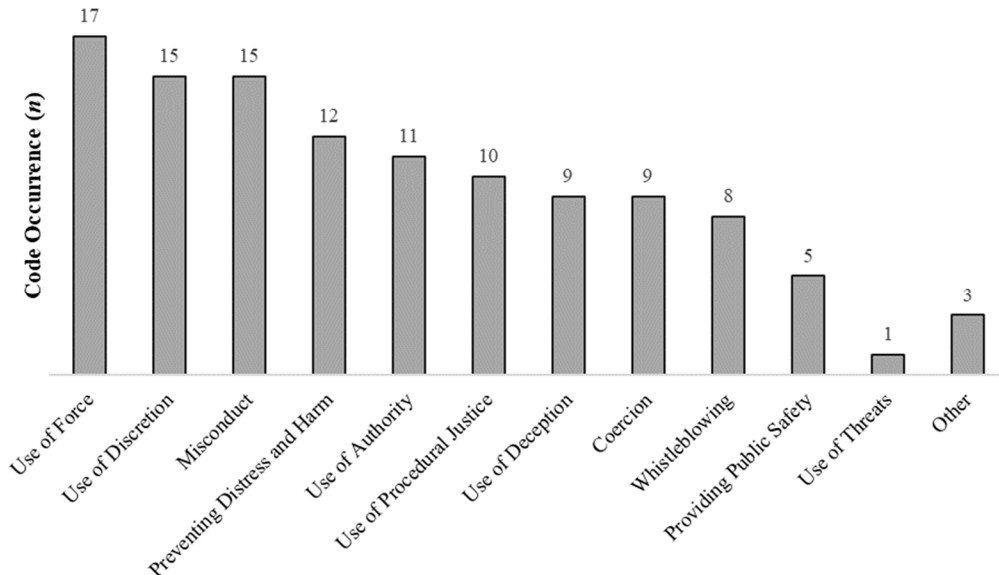

**Figure 6.** Count of deed nodes in the literature sample.

The most common consequence nodes were crime reduction (20%), unfavorable perception of police (19%), public trauma (13%), and moral injury (13%). Other consequences such as compliance (7%), favorable perception of police (6%), and crime increase (6%) were mentioned less frequently in the literature sample.

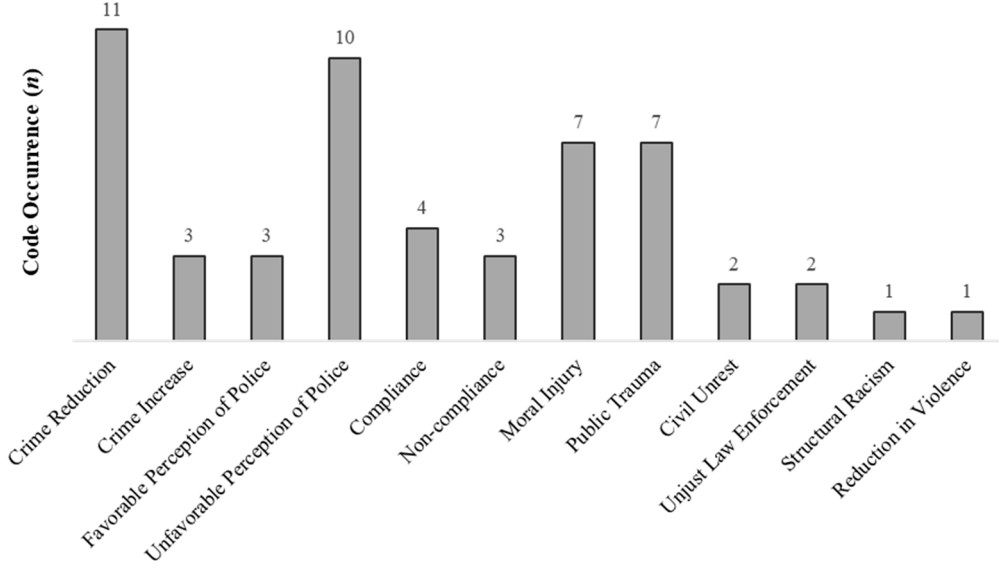

**Figure 7.** Count of consequence nodes in the literature sample.

**Table 3.** Agent, deed, and consequence domains with nodes and characteristic quotes from the sample illustrative of the socio-moral dimensions of policing, key virtues of law enforcement agents or police officers, and significant anticipated outcomes of decision-making in policing.

| Domain | Code | Excerpt | Source |
|---|---|---|---|
| agent | integrity | *"Research interest in police integrity has been growing in recent years [ … ]. However, the majority of these studies focus on police misconduct [ … ], which is only one perspective on integrity".* | [32] (p. 102) |
| agent | loyalty [1] | *"The police exposure to immediate threats and danger and a sense of insufficient skills to handle them lead to a collective ambition to survive the daily work and attempt to preserve one's self-esteem and pride. Building togetherness, loyalty, oneness, and a close identity with colleagues in the department are strategies for survival. Therefore, police departments are generally described as closed systems in which an esprit de corps develops [ … ]. Traditionally, police officers see themselves as professionally trained 'warriors' in a closed society, aiming to protect the 'good' citizen from the 'bad.' This warrior mentality forms the basis of their moral choices [ … ]".* | [28] (p. 36) |
| deed | use of force | *"Police wield an unrivaled power in society: The authority to use force in the interest of upholding state and federal laws. Although police rarely invoke this authority [ … ], their use of force is still a central concern to the public, a fact demonstrated, in part, by public demands for BWCs [body-worn cameras]. However, what constitutes acceptable or unacceptable force is not always clearly demarcated. The fundamental challenge is that police use of force is evaluated using a reasonableness standard: Would a reasonable officer have chosen the same course of action under the same circumstances [ … ]"?* | [33] (p. 734) |
| deed | use of deception | *"Significant ethical issues have long surrounded the use of deception in police operations. Whether discussed from a sociological [ … ], criminological [ … ], or philosophical [ … ] perspective, all allow those police at times need to use deception to be effective. Each also underscores the importance of a critical thinking and balanced approach in regard to two questions: (a) What are acceptable deceptive practices? (b) When are deceptive methods rather than more conventional methods of investigation justified"?* | [34] (p. 483) |
| consequence | moral injury [2] | *"Moral injury occurs when police officers perpetrate, fail to prevent, or bear witness to deaths or severe acts of violence that transgress deeply held moral beliefs (e.g., fatally shooting an allegedly armed criminal who is later proved to be unarmed)".* | [24] (p. 71) |
| consequence | unfavorable perception of police | *"In recent years, police use of force has been heavily criticized [ … ]. Videos of deadly force involving police and minorities have been prominent in the national news [ … ], and harsh criticism of police has consequently proliferated [ … ]. Some of this criticism is clearly warranted and is valuable as a catalyst for reforming unjust or insufficiently skilled practices; law enforcement agents are rightly held to a high standard of excellence, especially when employing violence [ … ]. However, some argue that a disproportionate amount of police violence in the media has resulted in public perceptions that the use of force by police is a common occurrence [ … ] when it is actually quite rare relative to the total number of police-citizen encounters [ … ]".* | [35] (p. 292) |

[1] As a function of survival affecting moral choices. [2] As a cause of moral injury.

### 3.2. Moral Psychology

Researchers who study how people make moral evaluations often distinguish two types of cognitive processes used in ethical decision-making [15,16]. A fast, unconscious, often emotion-driven system that draws from personal experience is contrasted with a slower, more deliberative, and analytical system that rationally balances benefits against costs among all available information. The fast, gut-level way of deciding is thought to have evolved earlier and to be the system that relies most on heuristics [11]. In the literature sample, fifteen percent of the articles mentioned moral intuition ($n = 8$, 15%), discussing experiential professionalism in policing whereby officers' actions are characterized by gut feelings, hunches, and intuition [36–38], officers making the right choice when there is no apparent good decision [24,39], and judgment of good and bad deeds as the product of intuitions [40]. We wish to consider the following excerpts about moral intuition from the perspective of police officers (agents) and applicability to police work in general.

**Moral Intuition (about agents):** "*The officers said that they often sensed quite quickly whether people were cooperative or not. Some of them said that they could spot a troublemaker from a long distance, 'Some people have an attitude problem'. One of the interviewed policemen talked about a 'gut feeling', saying, 'When I arrive at a scene where two people are fighting, I very quickly get a feeling that one of them is the idiot and the other one is OK. The idiot doesn't listen to me, but the other one does as he is told'. The officers said that they consistently tried to be 'good with the good guys and bad with the bad guys'.*" [37] (p. 77)

**Moral Intuition (general):** "*According to a Norwegian study, the internal status of Norwegian expert police officers is connected to 'experiential professionalism', which is 'characterized by gut feelings, hunches, intuition (rather than analysis), loyalty to colleagues, and attitudes aimed at crime control...' [ . . . ]. In a British setting, Loftus (2010) describes this 'sixth sense' as follows: '[ . . . ] the police learn to treat their geographical domain as a "territory of normal appearances". Their task is to become sensitive to those occasions when background expectancies are in variance.' The 'ecological' [ . . . ] rationality of experts, sensitive to what is present in and what is missing from a scenario, is intensified by a feeling of rightness [ . . . ]. This ability to discern nuances and its accompanying feeling of rightness may grow strong, almost incorrigible.*" [38] (p. 22)

*3.3. Structural and Normative Effects*

The structural and normative effects domain captures the essence of policing and social behavior and norms by uncovering the competing views of the structure-agency through various social factors and organizational perspectives within the law enforcement institution that influence the decision-making of police officers [41]. Furthermore, this domain illustrates the impacts of competing views on how society ethically evaluates police and the specific communities they serve. The categories of organizational (institutional) factors, social factors, and perceptions constituted the domain of structural and normative effects. From a broader perspective, some of the studies provide a constructive understanding of the role of social norms in the ethical decision-making process to help people. For example, one article defines the role of social norms as "morality related to the adherence of referential social norms" [42] (p. 343).

Organizational factors (*n* = 33, 61%) led to discussions of law enforcement at the micro or individual officer level, the meso-level, and the macro-level, which permeated across the domains and categories of the scoping review. Additionally, police subculture was often directly linked to organizational factors when discussed in the literature sample. For instance, one article suggested that friendship or team climate generally correlated with the motivation to blow the whistle [29]. Thirty-nine percent (*n* = 21) of the studies examined various aspects of social factors comprised of the following codes: sociocultural, socioeconomic, and sociopolitical variables in the context of law enforcement. In addition, the literature sample discussed the public perception of policing (*n* = 13, 24%) somewhat less frequently than the other two categories, with scholars in several instances discussing by what means the general public, the criminal justice system, and law enforcement evaluate police encounters to include the lethal use of force in accordance with social norms [35,43]. In combination with social factors and understanding police subculture, these evaluations offer insight into the decision-making calculus of law enforcement professionals. The following excerpts are illustrative of the interplay between organizational factors, the public perception of policing, and police subculture in the larger view of the structure-agency.

**Organizational Factors:** "*One promising avenue for reducing the influence of antagonistic emotions on police decision-making, then, is for agencies to strive to maintain an organizationally fair work environment for their officers. Indeed, some research suggests that officers who feel their supervisors are respectful, transparent, listen to their concerns, and otherwise treat them fairly tend to be less cynical and distressed, and in turn are less reliant on coercive force [ . . . ]*" [26] (p. 630)

**Public Perception of Policing:** *"The police face a dilemma in maintaining legitimacy: they must be perceived as effective in fighting crime and disorder, but they must also maintain standards of fairness [ . . . ]."* [37] (p. 784)

**Police Subculture:** *"While there is evidence that reform in Australia has generated some major successes, particularly in stopping organized protection rackets and substantially reducing police process corruption [ . . . ], Australian police departments continue to be subject to frequent conduct scandals. Recent high-profile cases in the media include racially motivated assaults and harassment; an escalation in fatal shootings; excessive tasering and assaults of suspects; and negligent responses to victims of crime, especially domestic violence [ . . . ]."* [44] (p. 371)

*3.4. Psychological Dimensions*

Attention to the psychological considerations sought to identify how emotions, biases, psychological disorders, examination of psychological theories, and the role of gender influence various aspects of ethical decision-making, specifically in high-stakes policing where officers often use intuitive cognitive processes to perform their duties in the field [16]. In terms of overall sample saturation, thirty percent (*n* = 16) of the articles discussed some aspect of emotional factors This scoping review further subdivided the category of emotions into the codes fear, shame, anger, anxiety, and disgust (see Table 4, emotional factors, nodes, for select code excerpts).

**Table 4.** Key emotions from the scoping review and their interplay into ethical decision-making in policing.

| Node (Code) | Excerpt | Source |
|---|---|---|
| fear | *"Discernment involves the ability to make judgments and reach decisions without being unduly influenced by extraneous considerations, fears, personal attachments, and the like".* | [14] (p. 880) |
| anxiety | *"For example, research has demonstrated that emotional exhaustion can lead to increased incidence rates of depression and anxiety among police officers [ . . . ]. Other research, not on a police sample, showed a relationship between anxiety and increases in unethical behavior [ . . . ]. Thus, although not yet empirically validated, police executives should be concerned that officers who experience work-related anxiety may be more prone to engage in unethical behavior".* | [30] (p. 2) |
| disgust (obesity) | *"Police officers were more likely to punish suspects who were obese than those who were not. This pattern occurred more for purity than for care crimes. These results illuminate the far-reaching implications of the amplification effect. When judging a person who belongs to a disgust-eliciting group, sensitivity to moral purity becomes heightened— good deeds are more highly praised, transgressions are more sharply criticized, and criminal behaviors are more readily punished".* | [40] (p. 2147) |
| shame and anger | *"Specifically, moral injury [ . . . ] is a condition that leaves officers with feelings of guilt, shame, anger, and betrayal... Such actions can leave officers questioning their moral values, which results in feelings of guilt and shame. Moral injury also occurs when officers feel betrayed by the unethical actions of trusted colleagues or supervisors. The sense of betrayal leads to feelings of anger. Moral injury and the emotions of those suffering from it may be key factors in helping to understand and to curtail the skyrocketing rate of police suicide".* | [45] (p. 2) |

The emotion fear was discussed the most often (*n* = 5, 9%), with concerns cited of fear interfering with job performance [26,37,45] and having the potential to influence discernment [14], and including the emotion in the appraisal of compassion [46]. The emotions of anxiety (*n* = 3, 6%), shame (*n* = 3, 6%), anger (*n* = 3, 6%), and disgust (*n* = 3, 6%) were marginally discussed in the literature sample. One article highlighted research that drew attention to emotional exhaustion as a mechanism for increasing anxiety among police officers and further showed through other research the potential consequences it could have on ethical behavior. The articles that examined disgust presented it as an influential emotion on moral judgment, classifying such behaviors eliciting the emotion as

offensive and creating negative biases and judgments towards people [40,47]. Separately, other articles related the emotions of shame and anger to moral injury and its impact on the mental health of police officers.

Biases are a critical psychological facet of understanding ethical decision-making. This scoping review focuses on any discussion of implicit biases in the literature sample based on how police officers can act on prejudices and stereotypes without conscious awareness factoring in the amount of discretion officers have and how it empowers them [48,49]. Furthermore, exploring implicit bias in law enforcement is integral to understanding its effects on police behaviors and its implications for intuitive cognitive processes in the context of split-second decision-making (i.e., use of force incidents). To further elaborate on implicit bias in the domain of law enforcement, there are efforts to distinguish implicit bias at the micro-level or individual officer level and relate instances of police brutality to the structure of law enforcement itself [50]. Overall, the literature sample infrequently mentions implicit bias (*n* = 9, 16%), with one article specifically relating implicit bias to procedural justice and moral judgment.

> **Implicit Bias:** "*Consciously or unconsciously, officers may convey their feelings or judgments about others in the degree to which they operate in a procedurally just (or unjust) fashion toward them. The degree of PJ [procedural justice] or injustice shown someone communicates a powerful symbolic message about the citizen's status or worth [ . . . ].*" [51] (p. 121)

This infrequency indicates a gap in applying factors of implicit bias to ethical decision-making in policing. It is worth noting separately that one article discusses artificial intelligence as a solution for addressing police brutality and implicit bias in American policing.

> **Implicit Bias (technology):** "*Given that American police have routinely (and justifiably) faced criticism for police brutality and implicit bias, particularly when dealing with minority communities, local police departments may feel pressure to seek automated solutions to issues like crowd control during protests or riots, by, for instance, programming rules of engagement into autonomous drones.*" [52] (p. 6)

Issues of psychological disorders (*n* = 4, 7%) led to discussions about post-traumatic stress disorder (PTSD) and suicide as the result of moral injury and suffering in police work [24,30], the psychological impacts that destructive obedience can have on policing behaviors centered on Milgram's paradigm [53], and the transition for military veterans who have PTSD, which creates the challenge of adapting to civilian settings, creating concerns for law enforcement employment [52].

> **Post-traumatic Stress Disorder (PTSD):** "*Papazoglou et al. [ . . . ] showed a relationship between moral injury [ . . . ] and incidence rates of post-traumatic stress disorder (PTSD). That connection can occur when the moral injury resulted from an incident or incidents in which the officer was exposed in some way to death, threatened death, actual or threatened serious injury, or actual or threatened sexual violence [ . . . ].*" [30] (p. 2)

A number of theories in psychology received both explicit and implicit mention throughout the literature sample, such as Bandura's work on moral disengagement (*n* = 15, 27%), social identity theory (*n* = 7, 13%), the interpersonal trust model (*n* = 7, 13%), and social learning theory (*n* =1, 2%). These psychological theories facilitate understanding the role of affect, dehumanization [54], and social and cultural factors in applying ethical decision-making to policing.

> **Moral Disengagement:** "*Perhaps the theory that best explains how policing fosters unethical decision-making is moral disengagement [ . . . ], which describes eight mechanisms whereby individuals are disinhibited from acting unethically. Each of the eight mechanisms of moral disengagement occur during routine police work [ . . . ]. Two of the mechanisms, dehumanization and attribution of blame, are particularly relevant when it comes to officers engaging in unethical behavior that will directly impact their wellness.*" [45] (p. 2)

> **Social Identity Theory:** *"This focus ignores the internal relationships among law enforcement personnel, where arrests function as an object of exchange and a medium of social connection. Existing research neither describes why officers desire to make arrests [ . . . ], nor explains the nature of the struggle between deputies to determine who will claim an arrest as his or her own."* [55] (p. 109)

The role of gender was discussed relatively infrequently (*n* = 7, 13%) and led to discussions of gender differences in compassion and empathy for biologically identifying males and females [46], differences in capabilities to perform police duties and use of force between biologically identifying males and females [35,44,56], understanding of gendered roles on policy and social norms [42,57], and the effect of gender on procedural justice in police and citizen encounters [51]. The steady increase of female recruitment into law enforcement further complicates the job complexities surrounding policing but demonstrates the need to understand how females who may be less comfortable with extremely forceful behaviors are able to better incorporate empathy and negotiation skills when having to make decisions [56].

> **Role of Gender (civilians):** *"Males may be perceived as more threatening, females as more submissive and in need of protection [ . . . ]. Males may also be viewed as less deserving and consequently receive less PJ. A bad reputation with the police is expected to reduce deservingness and police-provided PJ. Police researchers have not found strong or consistent effects of social status on police coercive practices [ . . . ], but recent meta-analyses have found that minority race/ethnicity significantly increases the risk of arrest, and one study finds that males are at significantly greater risk, while the effects of age are not significant [ . . . ]."* [51] (p. 122)

> **Role of Gender (officers):** *"Following women's entrance into police patrol in the 1970s, most research on female officers' capability to perform police duties has questioned their ability to maintain the typically masculine police qualities of physical aggression and force necessary to maintain police authority. Based on the traditional policing view, opponents of women police officers (namely, male officers) have argued vigorously that women cannot perform the job as well as men, due to their lack of physical strength and inability to maintain an authoritative presence [ . . . ]."* [56] (p. 426)

### 3.5. Gaps in Police Training and Policy

Gaps in police training and policy were identified throughout the literature sample. Over one-third of the literature sample (*n* = 20, 37%) mentioned training, and forty-eight percent (*n* = 26, 48%) of the sample mentioned policy implications. Training referenced recommendations for the need for police ethics training (*n* = 8, 15%), with mention of ethical decision-making training in three articles [26,43,46], and emphasizing the importance of moral reasoning in two articles [32,43]. One article emphasizes the need for "periodic training exercises that require officers to make quick decisions while in a state of heightened fear may improve officer decision-making in real-world encounters" [26] (p. 22). Separately, four articles acknowledged that recruit training is important to introduce recruits to procedural justice [51], that a training environment is conducive to developing community-youth communication skills [58], that emphasis is placed on ethics training in the academy [46], and that moral reasoning is important in both the training environment and the field [32]. Separately, one article identifies the use of force and assessing situations correctly as a training gap, based on police officer feedback expressing self-confidence issues [59].

> **Training Gaps (recruits/cadets):** *"Training could focus on police recruits before they have been exposed to the negative communications rituals of the street to teach them more sophisticated techniques of behavior management that are better suited to developing long-term relationships with youths. As community policing moves police into the "same cop, same beat" model, these strategies will be of critical importance to police safety and to the success of the community policing movement"* [58] (p. 40)

**Training Gaps (officers):** "*Although in many situations fear is warranted and even advantageous, officers who experience fear in response to civilians with bad attitudes may also be more likely to escalate the situation (e.g., by taking an aggressive stance, or drawing and pointing their firearm). Elevated fear may also be associated with perceptual distortions [ . . . ], which could result in mistake-of-fact shootings [ . . . ]. Periodic training exercises that require officers to make quick decisions while in a state of heightened fear may improve officer decision-making in real world encounters. More broadly, a better understanding of how civilian demeanor interacts with the situational context to influence fear among officers is important for efforts to reduce overreactions and mistakes that may result from increased fearfulness and that may be especially likely in certain environments*" [26] (p. 22)

*3.6. Interventions*

The literature sample in this scoping review narrowly focuses on interventions in policing (*n* = 6, 11%). Three articles explicitly discuss intervention strategies in detail, such as peer support mechanisms and family engagement [30], while the other three articles mention the importance of interventions to support officers with potential benefits mitigating misconduct risk [27], and addressing moral injury [24], as well as using White privilege interventions to raise awareness of systematic racial inequality in the United States [60].

**Interventions (race):** "*When making judgments about others, however, White privilege lessons may not trigger these self-preservation motivations and, instead, may simply highlight a system of racial inequality. And, in such a context, we find that White privilege lessons lead non-Black people—regardless of their political orientation—to perceive more racism when a Black man is shot by police. Critically, these shifts in perceived racism also had important downstream consequences for perceptions of legal guilt.*" [60] (p. 7)

There is detailed discussion in one article on the employment of a values-based methodology as an intervention strategy to help evaluate decisions in uncertain situations and foster experiential learning of best practices in the profession until they become second nature in practice [38]. However, as described in the limitations, the values-based methodology could prove to be difficult to employ during stressful and dynamic events but asserts that conditioning could create a pool of intuitive expertise. In the other two articles written by the same authors, they assert that effective interventions are the tell-tale signs of a healthy organization, commitment to officer wellness and ethics serves as an organizational solution to moral risks, and initiatives to improve wellness should also focus on ethical decision-making [30,45].

**Interventions (wellness):** "*The POWER perspective of wellness, ethics, and resilience helps law enforcement agencies and police officers to view wellness in an integrated way. Officers cannot stay healthy without maintaining a steadfast commitment to their core values. Similarly, when officers violate their moral code, their wellness will suffer. Initiatives to improve wellness should concurrently focus on officers' ethical decision-making. Likewise, efforts to reduce officers' misconduct must also address their physical, cognitive, emotional, spiritual, and social wellness. Although law enforcement agencies play a critical role in steps to improve officers' wellness and ethics, it is ultimately up to each individual police officer to incorporate a comprehensive wellness and ethics program.*" [45] (p. 3)

## 4. Discussion—(Un)Ethical Decision-Making

In general, decision-making refers to how people evaluate several alternatives, combine judgments and preferences, and choose the most probable choice to achieve one or more goals or desirable outcomes [61,62]. Moreover, the public expects law enforcement officers to make morally correct decisions in the execution of their duties regardless of the circumstances [24,33]. For instance, what governs the decision-making process when

a police officer decides whether to squeeze the trigger of a firearm or instead reach for a less lethal means such as a taser? This scoping review examined the ethical components of decision-making in law enforcement in 54 studies that met the inclusion criteria and provides a comprehensive, interdisciplinary summary of the findings. Three general themes emerged from the data that reveal the interaction between the job complexities of policing (psychological and normative factors) and aspects of ethical decision-making: (a) socio-moral dimensions impact the job complexities of police work, (b) lethal means and moral injury influence intuitive and rational decision-making, and (c) police wellness and interventions are critical to sustaining police readiness.

*4.1. Socio-Moral Dimensions Impact the Job Complexities of Police Work*

The domain of normative ethics in this scoping review captures the importance of police ethics, which has become highly significant in law enforcement with increasing scrutiny applied to the split-second decisions police officers make, primarily involving the use of force in the context of their duties [14,24,32,63]. There is research that focuses on the premise that rational actors make deliberate ethical decisions [64]. However, the findings from this scoping review suggest that ethical decision-making in policing is often less conscious and more intuitive and automatic [37,63]. This suggestive view is not meant to discount the substantial body of research on moral philosophy and psychology that makes use of the contrast between emotional and rational (cognitive) processes. Furthermore, the literature primarily focuses on single ethical theories and does not holistically examine the impact of multiple ethical theories on policing, creating gaps and disparities in how police officers apply moral decision-making to navigate the grey areas of law enforcement and the complex life and death situations police may encounter. This creates difficulties when crafting policies for law enforcement and determining their implications. Do policies that provide prescriptions governing the behaviors of police officers follow a Kantian duty approach or are they based on consequentialist calculations or decision procedures based on virtues and values? Perhaps, consideration of all three ethical theories can inform the policy process as demonstrated by the scope of the literature.

From a normative standpoint, the behavior of law enforcement professionals entails the cultural and social anchoring of specific moral guidelines from the organizational level to the individual officer level [65,66], which forms a police subculture and underpins the social learning that takes place in policing. These moral microcosms create moral risks and can lead to an increase in police misconduct [32,67]. However, law enforcement organizations can change their culture to better address the different moral risks encountered by their officers. Therefore, it is imperative for policymakers and law enforcement agencies to understand the social forces and multi-level factors that influence the likelihood for officers in a given situation to make unethical decisions in the course of their work and to accept or condone the deviant behaviors of others [23,27,68]. The end-state would be to responsibly educate the general public in order to foster community relationship-building with law enforcement and to create informed policies built upon public consensus.

Separately, the evidence shows that the moral principle police officers choose to apply is at times based on their emotions when confronted with unfamiliar micro-dilemmas involving split-second decisions and the use of force or other types of interventions. In particular, fear, shame, anger, anxiety, and disgust impact police officers' ability to perform their jobs and make sound moral judgments [26,37,45]. Additionally, the evidence shows that their choice is usually influenced by biases or outside pressures, such as the desire to conform to a police subculture. Therefore, while we likely believe we approach ethical dilemmas logically and rationally, our moral reasoning is usually influenced by intuitive, emotional reactions. This review also acknowledges a connection between emotions, moral injury, and the resulting psychological disorders and their impacts on police officers.

*4.2. Lethal Means, Moral Injury, and Decision-Making*

The cumulative moral impacts and risks from lethal use of force incidences and other stressors of the profession for law enforcement are significant and create the potential for police officers to disengage morally [24,54], and thus increase the possibility of making unethical decisions, resulting in police misconduct together with unfavorable consequences for the public [30]. Police officers question their moral values and ability to make morally correct decisions consistently. Consequently, these morally challenging circumstances often lead to moral injury [69], contributing to the issues of police violence and affecting public trust and confidence, leading to non-compliance and unfavorable attitudes towards police officers [26].

The dehumanization of victims, moral justification, and victim minimization [54] are critical for understanding moral disengagement and unavoidable when examining unethical decision-making in policing [70]. Unethical decision-making requires a choice on the part of an individual to accept a less than moral action given specific circumstances. Furthermore, past actions of a police officer, ranging from the use of force during an arrest to the use of deception when interacting with a suspect, could support the build-up of emotional distancing without training, intervention, and following procedural justice. For instance, some studies suggest that social and cultural distancing could contribute to the Black community's increased risk for police violence compared to other races and ethnicities [36,60]. This connection facilitates deviant policing subcultures and environments ripe for police misconduct, creating a slippery slope and contributing to a rise in calls for enacting detrimental policies for law enforcement. More specifically, this theme, combined with categories of psychological dimensions and normative factors, enabled an account of the socio-moral dimensions of policing, the virtues of law enforcement professionals, and the outcomes when investigating the moral aspects of decision-making and uncovered gaps in police readiness and wellness.

*4.3. Police Readiness, Wellness, and Interventions*

The capacity for police officers to use moral decision-making to navigate the ambiguous areas of law enforcement and the complex life-and-death situations police may encounter is of utmost importance to their readiness, wellness, and ability to perform their police work. Nevertheless, a separate study noted that less than one percent (less than six hours) of basic police training is spent on ethics training, demonstrating a capacity deficit and creating unnecessary risk to the force and the populace [9]. Coupled with the findings from the review, numerous training gaps have been identified outside of recruit training and stress the importance of continuous police training that addresses the nuanced complexities of police work with an emphasis on moral disengagement and moral injury [26,43,46]. These nuances include the emotional aspects of policing, with police officers second-guessing their decision-making, and, in certain instances, are worsened by fear and consequently create confidence issues. Likewise, the emotions of shame and anger contribute to ethical fading [64], and lead to moral injury, negatively impacting the mental health of police officers and moral discernment in their daily duties.

In addition, there is an empirical gap in the research on moral injury in policing, necessitating empirical study on the effects of moral suffering on law enforcement professionals based on anecdotal evidence and psychological disorders [23]. The review reveals that the research lacks granularity on the specifics of interventions when applied to ethical decision-making. Future research could bridge this gap and introduce potential programs, for example, that focus on the facets of mindfulness [71], lowering stress, and providing anger management for police officers to foster ethical decision-making and decrease misconduct. Finally, the potential exists to mirror and collaborate with military and veteran studies to help create policy and intervention programs.

The implementation of interventions in policing is imperative to the overall readiness and wellness of the law enforcement institution and its public perception and ability to secure the community's trust [72]. We found that many studies examined the use of

force, deception, use of discretion, and use of procedural justice as critical police actions (behaviors) and, when combined with universal virtues and the anticipated outcomes (consequences), play a vital role in understanding the job complexities of police work. Furthermore, a correct understanding of the intuitive basis of moral judgment may be useful in helping police officers avoid costly mistakes and assist trainers in designing programs (and environments) to improve the quality of moral judgment and police behavior [11,13]. The agent, deed, and consequence intuitions that we identified in the literature scope that trigger distinctive decision-making processes help create the ethical foundation for modeling learning algorithms (knowledge, skills, and attributes) involved in policing and incorporating it into a more holistic training curriculum. Leveraging an experiential learning-inspiring approach that accounts for multi-level factors to help guide law enforcement toward meeting the universal values of integrity, honesty, and compassion could lead to more ethical behavior and mitigate the shortcomings between action and outcome-based representation approaches. Our argument here, and as exposed by the scoping review, is that current research and best practices theoretically possess an ethical blind spot that is complemented by a holistic approach inspired by understanding the socio-moral dimensions of policing.

Moreover, law enforcement agencies strive to recruit, hire, and train only those who demonstrate strong moral values before entering the police training programs. Recruit training is essential to introduce recruits (cadets) to procedural justice, build citizen-communication skills, and develop ethics training that creates favorable views of law enforcement, increases compliance, and enhances perceptions of procedural justice [46,51,73]. Nevertheless, even departments' best efforts will not prevent instances of police misconduct from garnering attention. Such incidents undermine public trust, jeopardize important investigations, and expose agencies to considerable liability. As indicated in the review, commitment to officer wellness and ethics from leadership is essential to creating organizational solutions to improve policing and overcoming obstacles, such as deviant subcultures that impede police readiness.

## 5. Limitations

There are some limitations to consider when interpreting the results of our scoping review. Primarily, the scoping review was limited to two databases and did not include a grey literature search. Although the search strategy resulted in 54 articles after the full-text screening for inclusion, it is feasible that our review is missing additional relevant studies from other scholarly databases, with the current review demonstrating an interdisciplinary gap in the field and creating value in pursuing full systematic reviews as future research.

Given that this review does not assess motivational factors and does not formally appraise the quality of evidence, there is the possibility that some articles could present a level of bias or quality issues. A strict peer-review screening criterion for inclusion into the review and author affiliations were examined to mitigate this limitation. An appraisal of each author for every article was undertaken, and ninety-four percent of the literature sample was comprised of articles written by authors with an academic affiliation ($n = 51$), creating a negligible effect of any potential biases from affiliation conflict of interests and influences. In addition, the scoping review does not synthesize the relative weight of evidence in favor of the effectiveness of specific ethical decision-making strategies and any law enforcement program or intervention. Finally, this review included studies only written in English as a final point when considering limitations. Case in point, several studies were excluded for language accessibility, which could have proved helpful in the qualitative analysis when considering various aspects of ethical decision-making in law enforcement.

## 6. Conclusions

Ethical decision-making plays an influential role in numerous professions, such as the healthcare industry, the criminal justice system, and the military. Police officers endure

the challenges of continuous low-stakes and high-stakes dilemmas where making a split-second decision could have far-reaching consequences. Moreover, ethical decision-making involving moral judgment in ambiguous and stressful situations represents an emerging task in maintaining the confidence and trust of the public in law enforcement professionals in an environment where increases in police violence have sparked protests and prompted calls for police reform. This scoping review is, to our knowledge, the first comprehensive, interdisciplinary summary of research on ethical decision-making in law enforcement. Furthermore, this review investigates the existing literature using the ADC model combined with psychological and normative factors to understand the socio-moral dimensions of policing, systematizing analysis of the virtues of law enforcement professionals, policing behaviors and interactions with the public, and determining what the outcomes are when investigating the moral aspects of decision-making.

Three overarching themes emerged from the review. First, the socio-moral dimensions impact the job complexities of police work when combined with the psychological dimensions and normative factors such as the effect police subculture has on the behaviors of individual officers. Furthermore, multi-level factors are significant in the construction of an integrative approach to help guide law enforcement toward meeting the universal values of integrity, honesty, and compassion that lead to more ethical behavior and ethical decision-making, creating accountability, fairness in policing, and building trust in the communities they serve. Secondly, lethal means and moral injury influence intuitive and rational decision-making in policing. Lastly, police readiness, wellness, and interventions are critical to enabling police officers to use ethical decision-making to navigate the grey areas of law enforcement and the unfamiliar micro-dilemmas police may encounter with a heightened sense of ethics, psychological stability, and sociocultural awareness. Together, these themes synthesized an account of the socio-moral dimensions of policing and their impact on ethical decision-making and uncovered gaps in police training and policy.

This scoping review establishes the conditions for conducting meta-analyses in the future and creates the foundation for developing variables for empirical research in the domain of law enforcement and other interdisciplinary cross-sectional studies investigating ethical decision-making. Additionally, this review offers potential insight for studying artificial intelligence and data-enabled technologies and their applications, as well as ethical challenges for society and law enforcement (cf. [74]). While many articles in this scoping review acknowledge significant gaps and challenges to ethical decision-making in law enforcement, it is also clear that the most effective approach to improving policing is continued democratic discourse and engagement between the law enforcement profession and the public. Finally, future research should look to expand on the results here. It is the hope of the authors to take this information in this scoping review coupled with further investigation to refine ethics training to promote police readiness and wellness and further develop intervention tools for police officers.

**Author Contributions:** Conceptualization, R.P.D. and V.D.; methodology, R.P.D. and V.D.; software, R.P.D.; validation, R.P.D.; formal analysis, R.P.D., E.E.E. and V.D.; investigation, R.P.D. and E.E.E.; resources, R.P.D. and E.E.E.; data curation, R.P.D. and E.E.E.; writing—original draft preparation, R.P.D.; writing—review and editing, E.E.E. and V.D.; visualization, R.P.D.; supervision, V.D.; project administration, V.D. All authors have read and agreed to the published version of the manuscript.

**Funding:** This research received no external funding.

**Institutional Review Board Statement:** Not applicable.

**Informed Consent Statement:** Not applicable.

**Data Availability Statement:** No new data were created for this study. Data sharing is not applicable to this article.

**Acknowledgments:** The authors would like to thank James Brunet and Sam Cacace for their critical feedback and helpful guidance. For their valuable discussion and feedback, the authors would also like to thank the members of the NeuroComputational Ethics Research Group at North Carolina State

University—Austin Berg, Nora Edgren, Hannah Harwick, Brook Ireland, Seth Kodikara, Michael Pflanzer, and Olivia Matshabane.

**Conflicts of Interest:** The authors declare no conflict of interest.

## Appendix A

Scoping reviews are knowledge synthesis studies that follow a methodological approach to map the research done on a specific or heterogeneous range of topics, identifying and potentially facilitating the development of concepts and theories [18]. Additionally, scoping reviews aim to identify any existing gaps in knowledge and discuss future implications and research, including the value of pursuing full systematic reviews. See Table A1 for a list of the articles (studies) used in the scoping review and Table A2 for an overview of domains, main categories, and codes used in this scoping review. The articles cited in the body of the scoping review are also listed in the reference section.

**Table A1.** List of articles (studies) used in the scoping review.

| Article Number | Author (s) | Article Title | Publication Year |
|---|---|---|---|
| 1 | Ams | Blurred lines: The convergence of military and civilian uses of AI & data use and its impact on liberal democracy. | 2021 |
| 2 | Baldry and Pagliaro | Helping victims of intimate partner violence: The influence of group norms among lay people and the police. | 2014 |
| 3 | Blumberg et al. | Impact of police academy training on recruits' integrity. | 2016 |
| 4 | Blumberg et al. | Organizational solutions to the moral risks of policing. | 2020 |
| 5 | Blumberg et al. | The Importance of WE in power: Integrating police wellness and ethics. | 2020 |
| 6 | Brunson and Pegram | "Kids do not so much make trouble, they are trouble": Police-youth relations | 2018 |
| 7 | Buvik | The hole in the doughnut: A study of police discretion in a nightlife setting. | 2016 |
| 8 | Campbell and Fehler-Cabral | Accountability, collaboration, and social change: Ethical tensions in an action research project to address untested sexual assault kits (SAKs). | 2017 |
| 9 | Campbell et al. | The determination of victim credibility by adult and juvenile sexual assault investigators. | 2015 |
| 10 | Celestin and Kruschke | Lay evaluations of police and civilian use of force: Action severity scales. | 2019 |
| 11 | Clavien et al. | Choosy moral punishers. | 2012 |
| 12 | Connors et al. | The Mr. Big technique on trial by jury. | 2018 |
| 13 | Cooley et al. | Liberals perceive more racism than conservatives when police shoot Black men-But, reading about White privilege increases perceived racism, and shifts attributions of guilt, regardless of political ideology. | 2019 |
| 14 | De Schrijver and Maesschalck | The development of moral reasoning skills in police recruits. | 2015 |
| 15 | Donahue and Felts | Police ethics: A critical perspective. | 1993 |
| 16 | Dunnighan and Norris | Some ethical dilemmas in the handling of police informers. | 1998 |
| 17 | Girodo | Undercover probes of police corruption: Risk factors in proactive internal affairs investigations. | 1998 |
| 18 | Griffin et al. | Personal infidelity and professional conduct in 4 settings. | 2019 |
| 19 | Guarino-Ghezzi and Carr | Juvenile offenders versus the police: A community dilemma. | 1996 |
| 20 | Hough et al. | Misconduct by police leaders in England and Wales: An exploratory study. | 2018 |
| 21 | Howes et al. | Forensic scientists' conclusions: How readable are they for non-scientist report-users? | 2013 |

<div align="center">

**Table A1.** *Cont.*

</div>

| Article Number | Author (s) | Article Title | Publication Year |
|---|---|---|---|
| 22 | Jackson and Bradford | Crime, policing and social order: On the expressive nature of public confidence in policing. | 2009 |
| 23 | Jacobs | (Un)Ethical behavior and performance appraisal: The role of affect, support, and organizational justice. | 2014 |
| 24 | Johnson | The Enforcement of morality: Law, policing and sexuality in New South Wales. | 2010 |
| 25 | Juujärvi | Care and justice in real-life moral reasoning. | 2005 |
| 26 | Juujärvi | The ethic of care development: A longitudinal study of moral reasoning among practical-nursing, social-work and law-enforcement students. | 2006 |
| 27 | Kellough and Wortley | Remand for plea: Bail decisions and plea bargaining as commensurate decisions. | 2002 |
| 28 | Klockars | Blue lies and police placebos. | 1984 |
| 29 | Lande and Mangels | The value of the arrest—The symbolic economy of policing. | 2017 |
| 30 | Leach et al. | 'Intuitive' lie detection of children's deception by law enforcement officials and university students. | 2004 |
| 31 | Lentz et al. | Compromised conscience: A scoping review of moral injury among firefighters, paramedics, and police officers. | 2021 |
| 32 | Masicampo et al. | Group-based discrimination in judgments of moral purity-related behaviors: Experimental and archival evidence | 2014 |
| 33 | Mastrofski et al. | Predicting procedural justice in police-citizen encounters. | 2016 |
| 34 | Mercadillo et al. | Police culture influences the brain function underlying compassion: A gender study. | 2015 |
| 35 | Nix et al. | Compliance, noncompliance, and the in-between: causal effects of civilian demeanor on police officers' cognitions and emotions. | 2019 |
| 36 | Monaghan | On enforcing unjust laws in a just society. | 2018 |
| 37 | Morrell and Brammer | Governance and virtue: The case of public order policing. | 2016 |
| 38 | Navarick | Historical psychology and the Milgram Paradigm: Tests of an experimentally derived model of defiance using accounts of massacres by Nazi Reserve Police Battalion 101 | 2012 |
| 39 | Noppe | Dealing with the authority to use force: Reflections of Belgian police officers. | 2020 |
| 40 | Norberg | Legislation vs. morality—a police officer's ethical dilemma. | 2013 |
| 41 | Oberweis and Musheno | Policing identities: Cop decision making and the constitution of citizens. | 1999 |
| 42 | Papazoglou et al. | Addressing moral suffering in police work: Theoretical conceptualization and counselling implications. | 2020 |
| 43 | Park and Blenkinsopp | Whistleblowing as planned behavior—A survey of South Korean police officers. | 2009 |
| 44 | Paulsen | A values-based methodology in policing. | 2019 |
| 45 | Pellander | "An acceptable marriage": Marriage migration and moral gatekeeping in Finland | 2015 |
| 46 | Porter and Prenzler | The code of silence and ethical perceptions. | 2016 |
| 47 | Rabe-Hemp | Female officers and the ethic of care: Does officer gender impact police behaviors? | 2008 |
| 48 | Reamer | A narrative on the witch-hunt narrative: The moral dimensions. | 2017 |

**Table A1.** *Cont.*

| Article Number | Author (s) | Article Title | Publication Year |
|---|---|---|---|
| 49 | Renauer and Covelli | Examining the relationship between police experiences and perceptions of police bias. | 2011 |
| 50 | Rothwell and Baldwin | Ethical Climate Theory, whistle-blowing, and the code of silence in police agencies in the State of Georgia. | 2007 |
| 51 | Saulnier et al. | The effects of body-worn camera footage and eyewitness race on jurors' perceptions of police use of force. | 2019 |
| 52 | Sunshine and Tyler | Moral solidarity, identification with the community, and the importance of procedural justice: The police as prototypical representatives of a group's moral values. | 2003 |
| 53 | Visu-Petra et al. | An investigation of antisocial attitudes, family background and moral reasoning in violent offenders and police students. | 2008 |
| 54 | Wu and Makin | The differential role of stress on police officers' perceptions of misconduct. | 2021 |

**Table A2.** Overview of domains, main categories, and nodes (codes) used for the qualitative analyses.

| Domain | Categories | Codes |
|---|---|---|
| Normative Ethics | Agent | Integrity<br>Fairness<br>Ability<br>Loyalty<br>Honesty<br>Compassion<br>Empathy<br>Corruption<br>Trustworthiness<br>Empathy<br>Benevolence<br>Malevolence<br>Discernment |
| | Deeds | Use of Force<br>Use of Discretion<br>Misconduct<br>Preventing Distress and Harm<br>Use of Authority<br>Use of Procedural Justice<br>Whistleblowing<br>Coercion<br>Use of Deception<br>Providing Public Safety<br>Use of Threats<br>Other |
| | Consequences | Crime Reduction<br>Crime Increase<br>Favorable Perception of Police<br>Unfavorable Perception of Police<br>Compliance<br>Non-compliance<br>Moral Injury<br>Public Trauma<br>Civil Unrest<br>Structural Racism<br>Unjust Law Enforcement<br>Reduction in Violence |

**Table A2.** *Cont.*

| Domain | Categories | Codes |
|---|---|---|
| | Ethical Theories | Virtual Ethics<br>Deontology<br>Consequentialism<br>Ethics of Care/Harm |
| Psychological Dimensions | Emotions | Fear<br>Disgust<br>Anger<br>Shame |
| | Gender Dynamics | Role of Gender |
| | Biases | Implicit (Cognitive) |
| | Psychological Disorders | Post-Traumatic Stress Disorder<br>Suicide<br>Destructive Disobedience |
| | Psychological Theories | Moral Disengagement<br>Interpersonal Trust Model<br>Social Identity Theory<br>Social Learning |
| | Organizational Factors | Micro-level<br>Macro-level<br>Meso-level<br>Police Subculture |
| | Social Factors | Socio-cultural<br>Socio-economic<br>Socio-political<br>Public Perception of Policing |
| Interventions | | Interventions |
| Gaps | | Gaps |

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
