# Peer review of "Ethical Decision-Making in Law Enforcement: A Scoping Review"

_psych, doi:10.3390/psych5020037_

Round 1
Reviewer 1 Report
This is a very well done and scholarly paper that approaches the subject of the ethics of policing in a unique way. The only quibble I would pose is that, given its very academic style, I am not sure how accessible it will be for police practitioners and policy makers who could benefit from its findings?
- The main question addressed by the research is the nature and extent of the research evidence and knowledge bearing on ethical decision-making in law enforcement.
- Consider the topic original or relevant in the field, It pulls together the most valid current evidence on an important but little studied topic.
- Compared with other published material, I know of no other meta-analyses or what the authors call scoping investigations on this particular topic.
- I have no suggestions with respect to the methodology which is both appropriate and original.
- The conclusions consistent with the evidence and arguments presented and they address the main question posed.
- The references appropriate.
- The tables and figures are very clear and quite self-explanatory.
Author Response
Author response: Thank you for your positive feedback and comments. We agree that there needs to be a balance when bridging the rigor-relevance gulf and it should be bridged. Our hope here is twofold. One, given the amount of academic rigor, our methodology and findings provide a foundation for continued and expanded interdisciplinary research, and two the discussion and conclusions serve as an accessible catalyst for positive change for police practitioners and policymakers.
Reviewer 2 Report
Excellent paper. I enjoyed reading it and found the findings useful.
1. The main question addressed by the research are there gaps in how ethics is implemented in policing.
2. The topic original and relevant in the field. It address a specific gap in the field. There are serious criticisms of policing behaviors and outcomes and, at the same time, there are problems for Police in addressing difficult moral situations. The paper explores why ethical policing practices have been and are problematic and where solutions may be found.
3. It is a critical systematic review of existing material.
4. Specific improvements that the authors should consider regarding the Methodology and further controls: Relevant grey literature - an issue they have acknowledged in their Limitations section.
5. The conclusions consistent with the evidence, arguments presented and They address the main question posed.
6. The references appropriate.
7. Table 2 is hard to read.
Two typos:
1. Line 61 'respectively' should probably be 'respectfully'
2. Line 171 has a duplicate line/phrase
Author Response
Reviewer 2.
- Excellent paper. I enjoyed reading it and found the findings useful.
Author response: We would like to express appreciation for the constructive feedback and comments.
Comments from the Reviewer 2 about table accessibility and grammar.
- Table 2 is hard to read.
Author response: We have adjusted the table design and added an additional footnote to facilitate reading of Table 2 (line 194).
Two typos:
- Line 61 'respectively' should probably be 'respectfully'
- Line 171 has a duplicate line/phrase
Author response: Thank you for pointing this out and we have made the necessary grammatical corrections.
Line 61 has been corrected and the revised text reads as follows:
“The tough on crime perspective is conducive to ethical and moral decisions that reflect the need to combat crime and punish criminals, not treat them respectfully.”
The duplicate line/phrase from Line 171 has been deleted and highlighted in the manuscript as a tracked change.